# Insc:LGN tetramers promote asymmetric divisions of mammary stem cells

Simone Culurgioni [1,4], Sara Mari[1], Paola Bonetti [2], Sara Gallini[1], Greta Bonetto[1], Martha Brennich [3], Adam Round[3], Francesco Nicassio [2] & Marina Mapelli[1]

Asymmetric cell divisions balance stem cell proliferation and differentiation to sustain tissue morphogenesis and homeostasis. During asymmetric divisions, fate determinants and niche contacts segregate unequally between daughters, but little is known on how this is achieved mechanistically. In *Drosophila* neuroblasts and murine mammary stem cells, the association of the spindle orientation protein LGN with the stem cell adaptor Inscuteable has been connected to asymmetry. Here we report the crystal structure of *Drosophila* LGN in complex with the asymmetric domain of Inscuteable, which reveals a tetrameric arrangement of intertwined molecules. We show that Insc:LGN tetramers constitute stable cores of Par3–Insc-LGN-G$\alpha$i$^{GDP}$ complexes, which cannot be dissociated by NuMA. In mammary stem cells, the asymmetric domain of Insc bound to LGN:G$\alpha$i$^{GDP}$ suffices to drive asymmetric fate, and reverts aberrant symmetric divisions induced by p53 loss. We suggest a novel role for the Insc-bound pool of LGN acting independently of microtubule motors to promote asymmetric fate specification.

[1] Department of Experimental Oncology, European Institute of Oncology, Via Adamello 16, 20139 Milano, Italy. [2] Center for Genomic Science of IIT@SEMM, Istituto Italiano di Tecnologia (IIT), 20139 Milan, Italy. [3] European Molecular Biology Laboratory, Grenoble Outstation, Grenoble 38000, France. [4] Diamond Light Source Ltd., Diamond House, Harwell Science and Innovation Campus, Chilton, Didcot, Oxfordshire OX11 0DE, UK. These authors contributed equally: Simone Culurgioni, Sara Mari. Correspondence and requests for materials should be addressed to S.C. (email: simone.culurgioni@diamond.ac.uk) or to M.M. (email: marina.mapelli@ieo.it)

Stem cells have the remarkable ability to self-renew, meaning that upon asymmetric divisions they generate one daughter endowed with the same stem-like property of the mother, and another one prone to differentiate. Asymmetric cell divisions (ACDs in the following) are attained by unequal segregation of cell fate defining components, and by differential positioning of siblings within the tissue. Converging evidence revealed that in several stem cell systems, only daughters retaining contact to specialized microenvironment called *niches* maintain stemness[1]. Mechanistically, ACDs require the coordination of the division plane, and hence the mitotic spindle axis, with polarized cortical domains. Basic mechanism of spindle coupling to cortical polarity in ACDs have been elucidated in *Drosophila* neuroblasts and Sensory-Organ-Precursors[2], and found conserved in vertebrate stem cell systems including skin[3,4] and neural progenitors[5,6], and more recently murine mammary cap cells[7].

In epithelial stem cells, apico-basal polarity is established by enrichment of the polarity proteins Par3:Par6:aPKC at the apical site, which are able to recruit at the apical membrane an adaptor named Inscuteable (Insc). Insc was first identified in larval *Drosophila* neuroblasts as a partner of Par3 (Bazooka in flies), and later shown to bind the switch protein Pins (the fly orthologue of LGN, referred to as dLGN in the following)[8]. Mammalian homologs of fly Insc endowed with similar properties have been discovered in mouse developing skin[3], radial glia[9], and mammary stem cells[7], and are required for the correct execution of oriented ACDs. LGN in turns binds directly to the Dynein-associated protein NuMA with its tetratrico-peptide repeat (TPR) domain (hereon LGN$^{TPR}$) and to multiple Gαi subunits of heterotrimeric G-proteins, whose myristoyl group inserts into the lipid bilayer. During ACDs, the minus-end directed movement of Dynein engaged at the apical membrane with NuMA and LGN results in pulling forces on astral microtubules able to align the mitotic spindle along to the apico-basal polarity axis. Based on its interaction with both Par3 and LGN, and its involvement in oriented ACDs, Insc has long been considered the molecular connection between the polarity proteins Par3:Par6:aPKC and the spindle tethering machinery assembled on Dynein-NuMA:LGN: Gαi. Recent evidence indicate that Insc and NuMA are mutually exclusive interactors of LGN[10,11] raising the question as to how molecularly Insc works in ACDs.

The mammary gland is a branched ductal system consisting of a luminal epithelial layer surrounded by a myoepithelial contractile layer, embedded in a stromal matrix. In mammals, mammary gland development peaks at puberty and is driven by terminal end bud structures (TEBs) that forms at tips of the growing ducts and proliferate rapidly into the fat pad. TEBs in turn are composed by an outer basal layer, and multiple inner layers of luminal epithelial body cells. The regeneration cycles observed in mammary glands at pregnancies are sustained by mammary stem cells (MaSCs) residing in the TEBs[12], which have been shown to self-renew in an Insc-dependent manner[7,13]. More specifically, in mice during puberty, cap cells undergo mitosis with NuMA and LGN crescents polarized above one of the spindle poles[7,14]. Interestingly, although the existence of multipotent mammary stem cells in adult mice seems unlikely[15], it has been shown that mammary epithelial cells isolated from murine mammary gland gain stemness in vitro and exhibit pluripotency. Insc has been reported to promote ACDs also in *Drosophila* neuroblasts, murine skin progenitors and neural stem cells[6,16–18].

Structural studies revealed that Insc codes for a conserved 35-residue peptide (Insc$^{PEPT}$ hereon) binding to the N-terminal TPR domain of LGN/dLGN with nanomolar affinity[10,11,19]. Beside Insc$^{PEPT}$, vertebrate and invertebrate orthologues of Insc share poor sequence homology. Both proteins are predicted to contain a helical rich region downstream of the LGN-binding peptide,

which in the case of the *Drosophila* Insc is sufficient to recapitulate its localization and functions in asymmetric divisions of neuroblast (hence termed Insc$^{ASYM}$)[20,21]. No structure-function analysis is to date available for human Insc besides N-terminal truncations shown to impair binding to LGN[9]. Intriguingly, in vitro human Insc has been reported to interact with the PDZ domains of Par3 via a conserved C-terminal motif[22,23], outside of the central helical fragment. To understand the molecular basis of Insc functioning, we set out to study the organizational principles of the Par3–Insc–LGN–Gαi interaction, and their impact on murine mammary stem cell asymmetric divisions. Our studies revealed that LGN and Insc form stable tetramers at the core of Par3–Insc–LGN–Gαi macromolecular assemblies, which are required for asymmetric fate acquisition of MaSCs, and do not exchange LGN molecules with microtubule motors.

## Results

**Molecular architecture of the dLGN$^{TPR}$:Insc$^{ASYM}$ tetramer.** Previous structural studies have revealed that the flexible Insc$^{PEPT}$ binds to the inner side of the TPR α-solenoid of LGN/dLGN with an extended conformation[10,19]. To determine the atomic model of the whole Insc$^{ASYM}$, we took advantage of a discistronic pGEX vector co-expressing GST-dLGN$^{25–406}$ with the *Drosophila* Insc$^{ASYM}$ construct encompassing residues 283–623 (Supplementary Fig. 1a). The structure of dLGN$^{TPR}$:Insc$^{ASYM}$ was solved by molecular replacement using the coordinates of dLGN$^{TPR}$: Insc$^{PEPT}$ as a search model. Multi-crystal averaging was employed to obtain an interpretable electron density for manual model building of the Insc$^{ASYM}$ domain. The final model was refined at 3.4 Å resolution to an $R_{free}$ of 25.0% and $R_{work}$ of 20.9% with good stereochemistry (Table 1 and Supplementary Fig. 1b, c). It consists of residues 41–386 of dLGN and residues 302–595 of Insc$^{ASYM}$, thus lacking about 20 residues preceding the Insc$^{PEPT}$ stretch.

### Table 1 Data collection and refinement statistics

| Pins$^{TPR}$-Insc$^{ASYM}$ | Dataset 1 | Dataset 2 |
|---|---|---|
| *Data collection* | | |
| Space group | P2$_1$2$_1$2$_1$ | P2$_1$2$_1$2$_1$ |
| Cell dimensions | | |
| $a$, $b$, $c$ (Å) | 128.19, 212.58, 280.73 | 127.08, 214.07, 279.68 |
| $\alpha$, $\beta$, $\gamma$ (°) | 90° 90° 90° | 90° 90° 90° |
| Resolution (Å) | 75.6–3.4 (3.52–3.4)* | 70.6–4.0 (4.1–4.0)* |
| $R_{merge}$ | 0.2328 (1.422) | 0.3384 (1.896) |
| $I/\sigma I$ | 12.7 (2.15) | 6.21 (1.23) |
| Completeness (%) | 99.9 (99.98) | 99.8 (98.9) |
| Redundancy | 14.9 (14.1) | 6.7 (7.0) |
| *Refinement* | | |
| Resolution (Å) | 75.6–3.4 | |
| Reflections used | 105843 | |
| $R_{work}/R_{free}$ | 0.209/0.249 | |
| No. atoms | 38983 | |
| Protein | 38979 | |
| Ligand/ion | 0 | |
| Water | 4 | |
| *B-factors* | | |
| Protein | 78.0 | |
| Ligand/ion | 0 | |
| Water | 30 | |
| R.m.s deviations | | |
| Bond lengths (Å) | 0.003 | |
| Bond angles (°) | 0.53 | |

*Values in parentheses are for highest-resolution shell

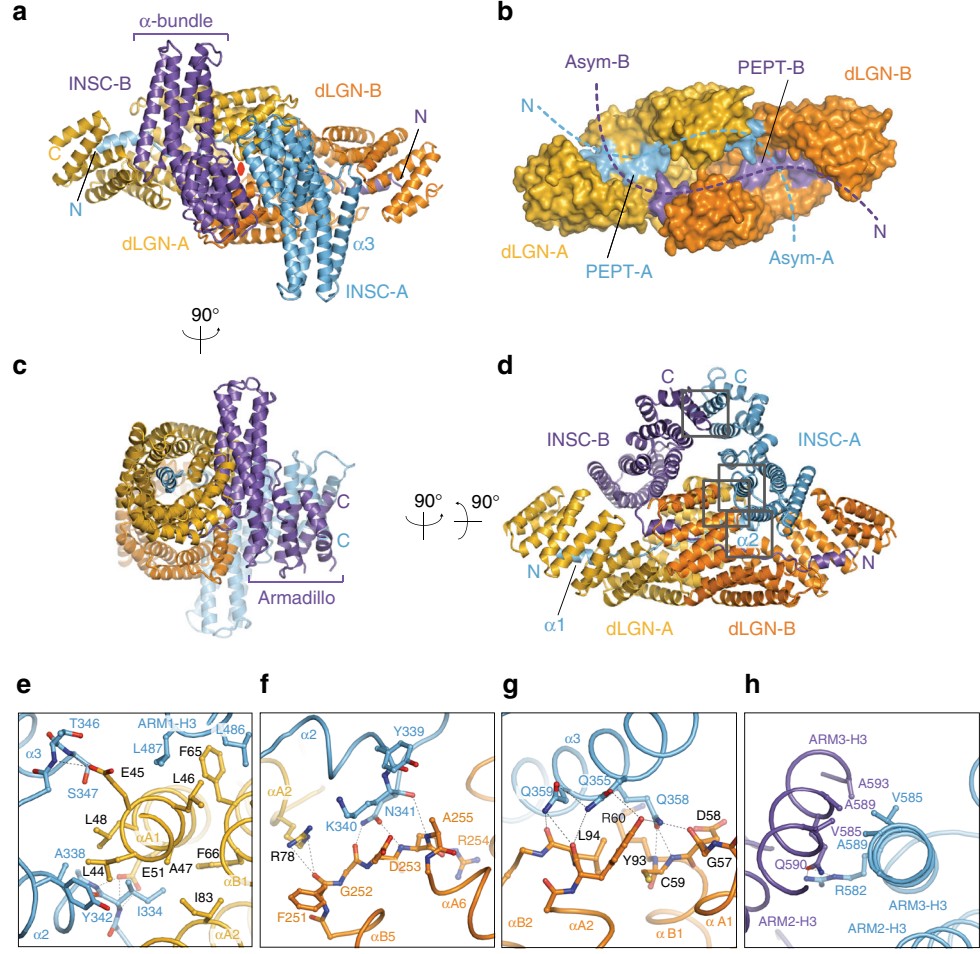

**Fig. 1** Structure of the dLGN$^{TPR}$:Insc$^{ASYM}$ complex. **a–d** Cartoon representation of the dLGN$^{TPR}$:Insc$^{ASYM}$ tetramer in three orthogonal orientations. The two protomers of dLGN$^{TPR}$ are colored gold and orange, while the two Insc$^{ASYM}$ are in cyan and purple. The two-fold symmetry axis of the tetramer is marked as a red oval in **a**. **b** Surface representation of the arrangement of the dLGN$^{TPR}$ domains with the N-terminal fragments of Insc$^{ASYM}$-labeled PEPT (Insc$^{PEPT}$ in the text) in the same orientation and colors as in **a**. **e–h** Close-up views of the interaction of Insc-$A$ with the other subunits of the tetramer from the N terminus to the end of the polypeptide chain. The positions of the individual close-up views within the complex are indicated in **d**. Interacting residues at the interfaces are depicted in balls-and-sticks

The most striking feature of the dLGN$^{TPR}$:Insc$^{ASYM}$ structure lies in its hetero-tetrameric arrangement (Fig. 1a), whereby each Insc chain contacts one dLGN$^{TPR}$ with the elongated Insc$^{PEPT}$ stretch and the second dLGN$^{TPR}$ with a four-helix bundle, in a sort of intimate domain-swap (Fig. 1b). The C-terminal portion of the Insc$^{ASYM}$ chains protrudes away from the TPR domains, and folds in Armadillo units, packing to one another via the last helices of the repeat (Fig. 1c, d). In the tetramer, the two dLGN$^{TPR}$ molecules form homotypic head-to-head interactions by shape complementarity mediated by the linkers joining TPR3-4-5 of the two subunits. Topologically, this packing of the TPR domains does not prolong the super-helical array of consecutive TPR motifs, but rather generate a cylindrical scaffold entirely closed on one side (Fig. 1b). The two Insc$^{PEPT}$ fragments run antiparallel to one another lining along the inner side the TPR channel, whose open side is occupied by the helical bundles of the Insc$^{ASYM}$ folds (Fig. 1c). This way, each Insc$^{ASYM}$ chain precludes solvent accessibility to the dLGN$^{TPR}$ molecule to which it is not in contact with its own Insc$^{PEPT}$. The extended tetramer interface generated by a non-crystallographic two-fold axis roughly passing through the helical domains of Insc$^{ASYM}$ buries a surface of about 8000 Å$^2$ (Fig. 1a). Interestingly, self-association of TPR domains has been reported for several components of the Anaphase-Promoting Complex (APC), the E3-ligase driving mitotic exit[24].

However, APC subunits homodimerize by hooking of the N-terminal TPRs, which engage in a tight clasp-like interaction[25], whereas dLGN$^{TPR}$ protomers are monomeric in isolation and dimerize only in the presence of Insc ligands longer than the peptide fragment (Supplementary Fig. 2).

Looking from the side of the TPR domains, the dLGN:Insc tetramer resembles a sailing boat (Fig. 1d), whose hull is formed by the TPR scaffolds and sails by the helical domains of Insc$^{ASYM}$. The symmetrical arrangement of the tetramer is reflected in equivalent interactions among the subunits. To facilitate the description we will name the dLGN and Insc couples interacting via the Insc$^{PEPT}$ as A and B. Several conserved hydrophobic and polar interactions between Insc$^{PEPT}$ and dLGN$^{TPR}$ account for the nanomolar binding affinity of the binary interaction[10,11,19]. The Insc$^{PEPT}$ stretch of Insc-$A^{ASYM}$ lines along the inner dLGN-$A^{TPR}$ surface as seen in the dLGN$^{TPR}$:Insc$^{PEPT}$ structure. Remarkably, in the tetramer, residues 334–342 of Insc-$A^{PEPT}$ fold into a second α2-helix that packs parallel to the αA1–αA2 helixes of dLGN-$A^{TPR}$ with primarily hydrophobic interactions (Fig. 1e and Supplementary Movie 1). This α2-helix of Insc-$A^{ASYM}$ is capped by TPR5-6 of dLGN-$B^{TPR}$, in an interaction contributed by hydrogen bonds between Lys340-Asn341$^{Insc-A}$ and Gly252-Asp253-Ala255$^{dLGN-B}$ (Fig. 1f). After wedging between the two TPR domains, the Insc-$A^{ASYM}$ chain undergoes

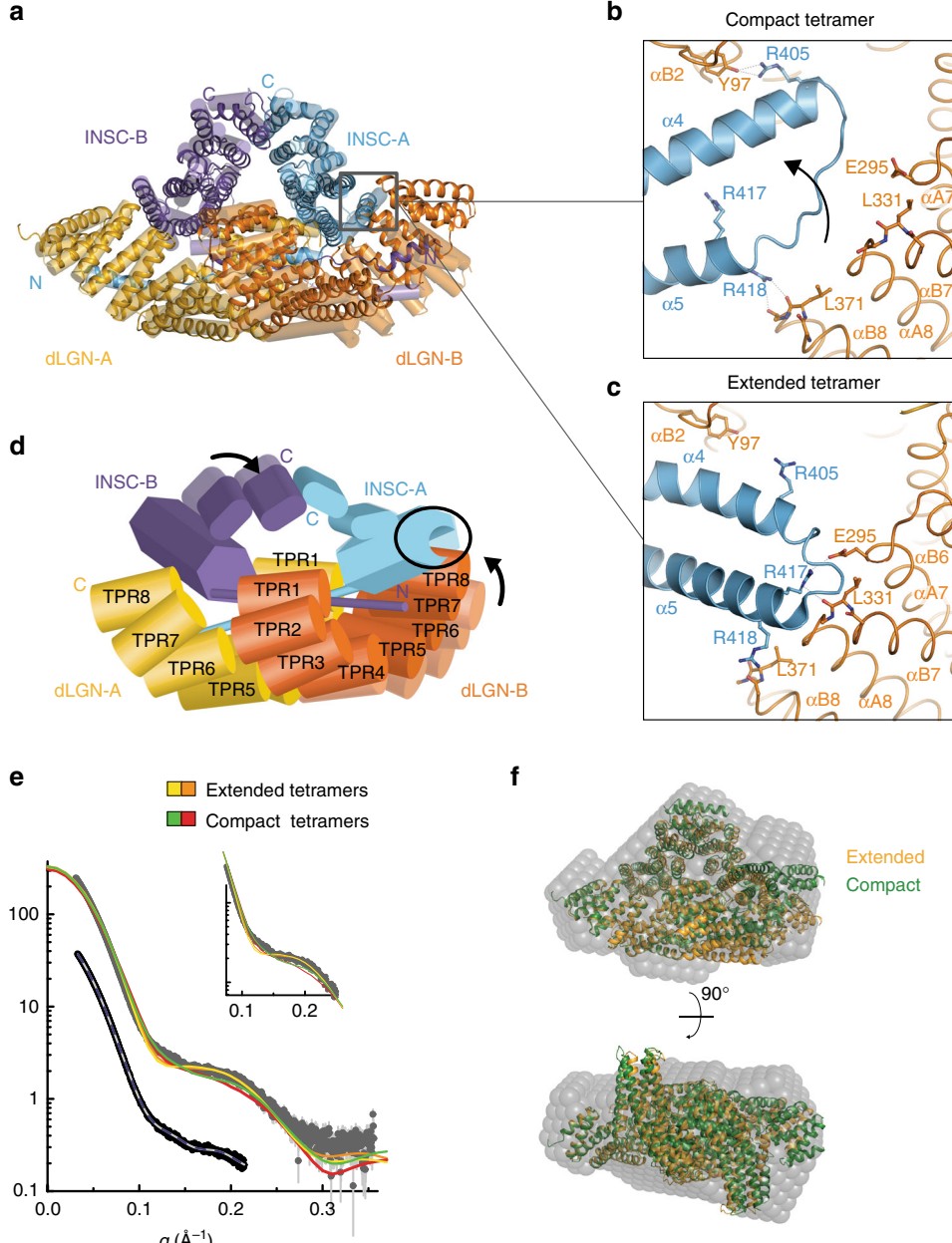

**Fig. 2** The dLGN[TPR]:Insc[ASYM] complex adopts two conformations. **a** Superposition of the two dLGN[TPR]:Insc[ASYM] conformers present in the a.s.u. The most compact conformer is shown as ribbon diagram, and the most extended is depicted in transparent cylindrical helices. Subunits are colored as in Fig. 1. **b, c** Zoomed views of the interfaces between the helical bundle of Insc-*A* and TPR7-8 of dLGN-*B* for the compact (top) and extended (bottom) tetramers. The rearrangement of the TPR repeats of dLGN-*B* in the compact conformation is accompanied by disruption of the initial turns of helix α5 of Insc-*A*. **d** Graphical summary of the conformational changes between the extended and the compact tetramers. **e** SAXS data for the dLGN[TPR]:Insc[ASYM] complex in comparison to the theoretical curves for the different configurations (upper part and inlay) and fits of ab initio models in P1 (gray) and P2 symmetry (blue). Idealized curves were obtained merging frames with matching radius of gyration and high similarity according to CORMAP[38,39]. **f** The ab initio model of the dLGN[TPR]:Insc[ASYM] complex in P1 symmetry (transparent gray spheres) matches well with both the extended (orange) and compact (green) tetramers

an abrupt kink to form the first helix of the bundle (the α3 helix of Insc[ASYM], Fig. 1a), running almost orthogonal to the dLGN-*A* TPR repeats. Compared to the extensive contacts engaging the Insc[PEPT], fewer specific interactions are found in the rest of the interface between Insc and dLGN molecules. A cluster of Gln residues in the middle of the α3 helix of Insc-*A*[ASYM] packs against the intra-repeat loops of the TPR1-2[dLGN-B] making bidentate hydrogen bonds with main chain atoms and staking interactions with Tyr93[dLGN-B] (Fig. 1g). These contacts create a topological closure of the TPR channel. In the center of the

tetramer hull, the symmetrical interface between dLGN-*A*[TPR] and dLGN-*B*[TPR] is contributed primarily by hydrogen bonds between main chain atoms of TPR turns, as exemplified at TPR4-5 (Supplementary Fig. 3a, b). The corresponding interface at the tip of the *sails* is maintained by pairing of the α3-helices of the third Armadillo repeats via hydrophobic interactions between Val585[Insc] and Ala589[Insc] of both chains, further strengthened by a polar interaction between Arg582[Insc-A] and Gln590[Insc-B] (Fig. 1h and Supplementary Movie 1). Thus, based on structural evidence, we can conclude that (1) the dLGN[TPR]:Insc[ASYM] is a

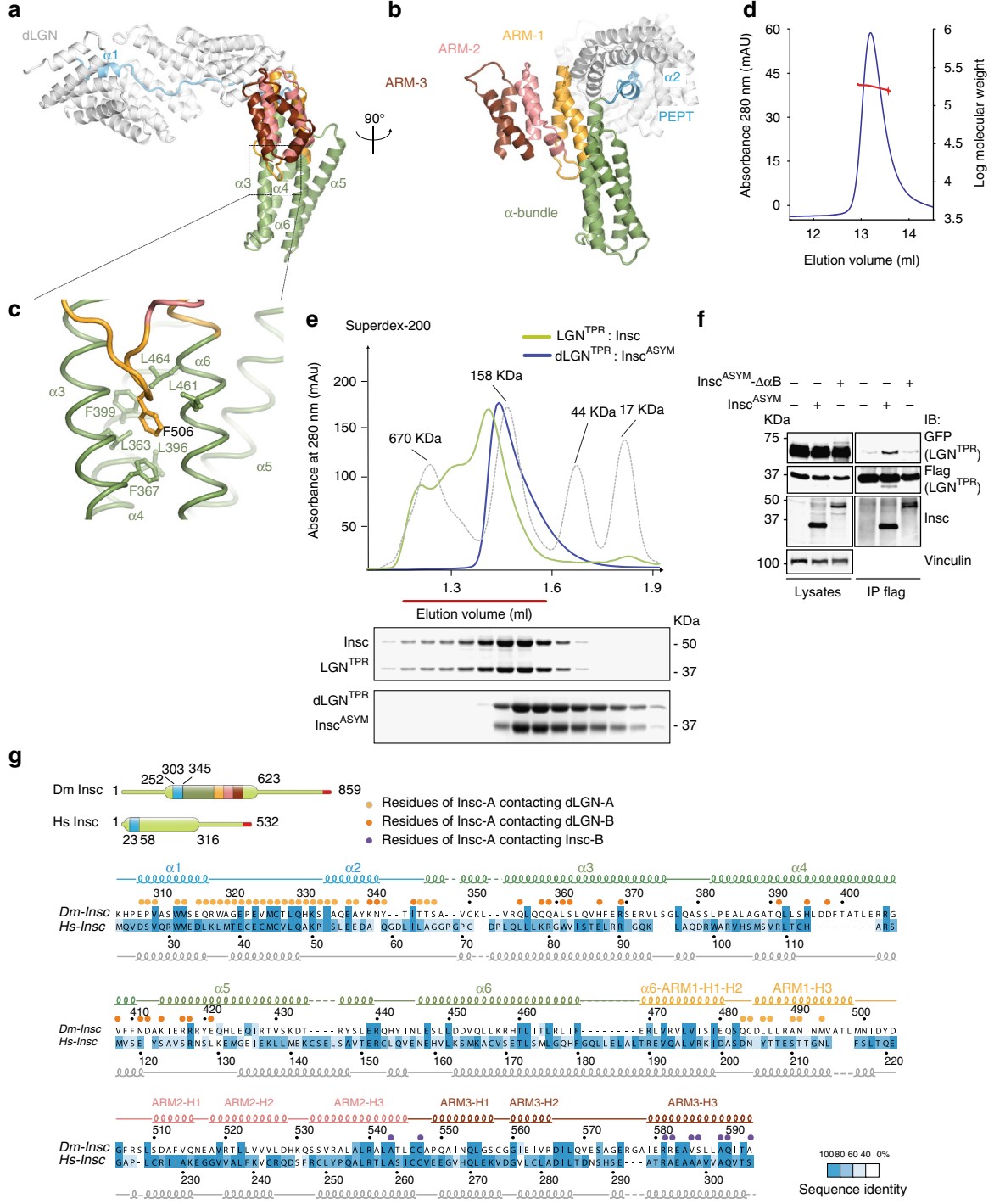

**Fig. 3** The topology of the asymmetric domain of Insc is evolutionary conserved. **a**, **b** Cartoon model of Insc[ASYM] bound to the inner groove of dLGN[TPR] (in white) at two orthogonal views. After the initial Insc[PEPT] stretch (in cyan), the Insc chain departs from the TPR α-solenoid forming a four-helix bundle (green). The C-terminal part of the last helix of the bundle kinks into two short helical fragments of the first non-canonical ARM repeat (gold), which is followed by two conventional ARM motifs adopting a triangular flat shape (pink and brown). **c** Zoom of the hydrophobic pocket in which Phe506[Insc] fits. **d** Static-Light-Scattering profile of the dLGN[TPR]:Insc[ASYM] tetramer showing an average molecular mass of about 180 kDa along the peak, as expected for a 2:2 tetramer. **e** SEC elution profiles of *Drosophila* dLGN[TPR]:Insc[ASYM] (blue trace) and human LGN[TPR]:Insc (green trace) with associated Coomassie-stained SDS–PAGE separation of peak fractions. The elution profile of globular markers is reported in a dashed gray line. Both complexes elute in fractions consistent with the theoretical molecular mass of 2:2 tetramers. **f** Insc[ASYM]-ΔαB assembles with LGN[TPR] in a 1:1 stoichiometry. HEK293T cells were transiently transfected with plasmids containing human GFP-LGN[TPR] and FLAG-LGN[TPR] (residues 1–350) alone or in combination with human Insc[ASYM] or Insc[ASYM]-ΔαB (lacking residues 62–191). After 48 h, cells lysates were immunoprecipitated (IPs) with anti-FLAG antibodies conjugated to sepharose beads, and immunoblotted (IB) with the indicated antibodies. FLAG-LGN[TPR]- was able to co-immunoprecipitate with GFP-LGN[TPR] only when bound to Insc[ASYM], but not to Insc[ASYM]-ΔαB. **g** Sequence alignment of Insc[ASYM] with secondary structure elements based on the crystallographic structure (for *Drosophila*) and in silico prediction (for *Homo sapiens*). Residues are colored according to the degree of conservation calculated on the basis of the alignment of Supplementary Fig. 4. Circles indicate residues of Insc[ASYM] in contact with the other three subunits of the tetramer

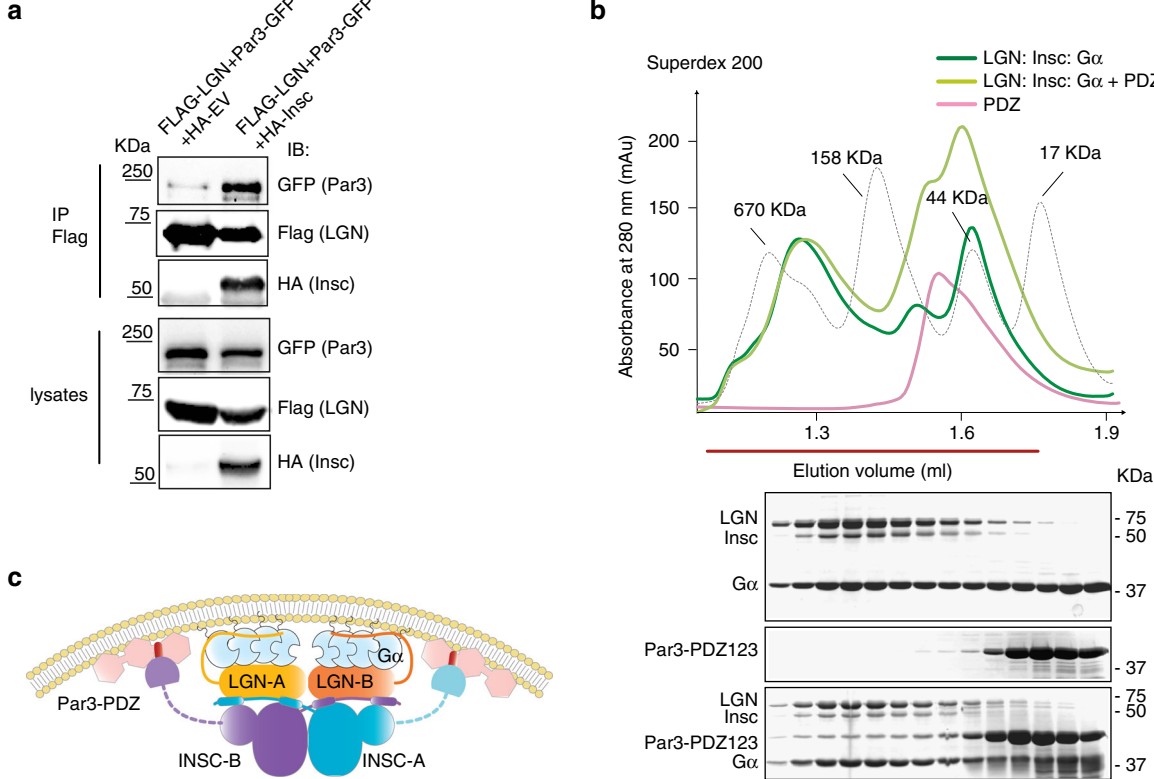

**Fig. 4** Conserved organizational principles of Par3:Insc:LGN:Gαi complexes. **a** Reconstitution of human Par3:Insc:LGN interaction in cells. HEK293T cells were transfected with Par3-GFP, FLAG-LGN and HA-Insc. Cell lysates were subjected to IP with α-FLAG antibody conjugated to sepharose beads. After washes, species on beads were analyzed by SDS–PAGE and immunoblotted as indicated in the figure. HA-empty vector was included as a negative control of the IP. **b** SEC elution profiles and SDS–PAGE separation of Insc:LGN:Gαi complexes reconstituted from expression in insect cells (dark green trace) and Par3$^{PDZ123}$ purified from bacterial sources (pink trace). When the Par3$^{PDZ123}$ is mixed in molar excess with Insc:LGN:Gαi, it coelutes with Insc:LGN:Gαi in a stoichiometry reflecting the amount of Insc in the complex (light green trace). **c** Summary of the interactions in the Par3:Insc:LGN:Gαi complex supported by our study. Stable Insc:LGN tetramers form the core of apically localized Par3:Insc:LGN:Gαi complexes, in which the C-terminal portion of the two Insc chains contacts directly the PDZ domains of Par3. Binding of each LGN-GoLoco to four Gαi$^{GDP}$ molecules stabilizes the association of the macromolecular assemblies to the plasma membrane

symmetrical tetramer of intertwined subunits, whose high affinity binding is dictated by extended interactions between the Insc$^{PEPT}$ and the TPR domains; (2) the helical portions of the Insc$^{ASYM}$ chains introduce into the tetramers a topological constrain preventing the dissociation of the complex.

**Conformational flexibility of the Pins:Insc tetramers.** Insc works in complexes localized in proximity of the plasma membrane, which is an inherently flexible structure. Consistent with this role, we found that the four independent copies of the tetramer present in the asymmetric unit display two slightly different conformations, one extended (described in the previous paragraph) and the other more compact. Comparison of the two classes of tetramers revealed the existence of a structurally invariant rigid body including one of the dLGN$^{TPR}$ domains (referred to as dLGN-$A^{TPR}$ in the following), and the cognate Insc-$A^{PEPT}$. Upon superposition on this rigid body, a major rearrangement at the C-terminal end of dLGN-$B^{TPR}$ becomes visible (Fig. 2a). The compaction of the tetramer hull is transmitted allosterically to the Insc-$A^{ASYM}$ domain, whose helical bundle is disrupted to accommodate TPR7-8 of dLGN-$B$ (Fig. 2b–d, Supplementary 3a–d and Supplementary Movie 2). To assess whether the structural variability observed in the crystal packing is preserved in solution, we performed SEC-SAXS analyses of the complex. The low-resolution ab initio shape determined from the experimental data provided a particle envelope

consistent with the tetramer (Fig. 2e–f and Supplementary Fig. 3f), with a radius of gyration $R_g$ of about 41 Å, and maximum particle size $D_{max}$ of 120 Å. Interestingly, the experimental data deviates substantially from the scattering pattern calculated from extended and compact tetramers individually (discrepancy $\chi^2$ in the order of 8.0), suggesting that in solution these assemblies are flexible and free to sample a conformational space including, but not limited, to the arrangements observed in the crystals. We conclude that the dLGN$^{TPR}$:Insc$^{ASYM}$ tetramer is endowed with an intrinsic structural plasticity mirroring the extended and compact organization of the crystallographic structures.

**Conserved topology of the Insc$^{ASYM}$ domain.** A great challenge in the stem cell field deals with the understanding of the molecular mechanism accounting for vertebrate asymmetric cell divisions. Human Insc was identified as a protein endowed with the same interaction properties and spindle orientation functions of fly Insc. Nonetheless invertebrate and vertebrate Insc orthologues share very poor sequence homology (Fig. 3g and Supplementary Fig. 4). To understand whether the architecture of the dLGN$^{TPR}$:Insc$^{ASYM}$ oligomers recapitulate the organizational principles of human counterparts, we carried out a comparative analysis of Insc in the two species, starting from the topology of the asymmetric domain. After the TPR-interacting peptide, *Drosophila* Insc$^{ASYM}$ forms an elongated helical bundle, followed by a globular domain of three Armadillo (ARM) repeats (Fig. 3a,

b). Each ARM unit consists of two short helices followed by a long one, designated H1, H2, and H3. In fly Insc$^{ASYM}$, the first ARM is contributed by the C-terminal half of helix-α6, which bends to conform to the orientation of ARM helices H1 and H2 (Fig. 3b). Insertion of Phe506$^{Insc}$ into a hydrophobic pocket organized by Leu and Phe residues of helices α-4 and α-6 of the bundle maintains the ARM domain orthogonal to the helical bundle (Fig. 3c). To define whether human Insc adopt the same structure, we compared its secondary structure prediction with the experimentally defined helical pattern of *Drosophila* Insc$^{A-SYM}$, and found that the fragment of about 300 residues after the human Insc$^{PEPT}$ is predicted to assume a helical conformation with a sequence of short and long elements consistent with the fly

Insc$^{ASYM}$ fold (Fig. 3g), suggesting that the two Insc orthologues fold similarly. We then asked whether human Insc could oligomerize with LGN$^{TPR}$. To this end, we purified to homogeneity the human LGN$^{TPR}$:Insc complex expressed recombinantly in insect cells from a dicistronic baculovirus vector. In line with the crystallographic and SAXS data, Static-Light-Scattering analysis confirmed that the *Drosophila* dLGN$^{TPR}$:Insc$^{ASYM}$ complex elutes from a size-exclusion column as a tetramer (Fig. 3d). When loaded on a size-exclusion column, the human LGN$^{TPR}$:Insc complex elutes slightly earlier than the dLGN$^{TPR}$:Insc$^{ASYM}$ tetramer (Fig. 3e), and before the 158 kDa molecular weight marker, indicating that its most abundant oligomeric state is a 2:2 heterotetramer. We reasoned that the difference in the elution profiles

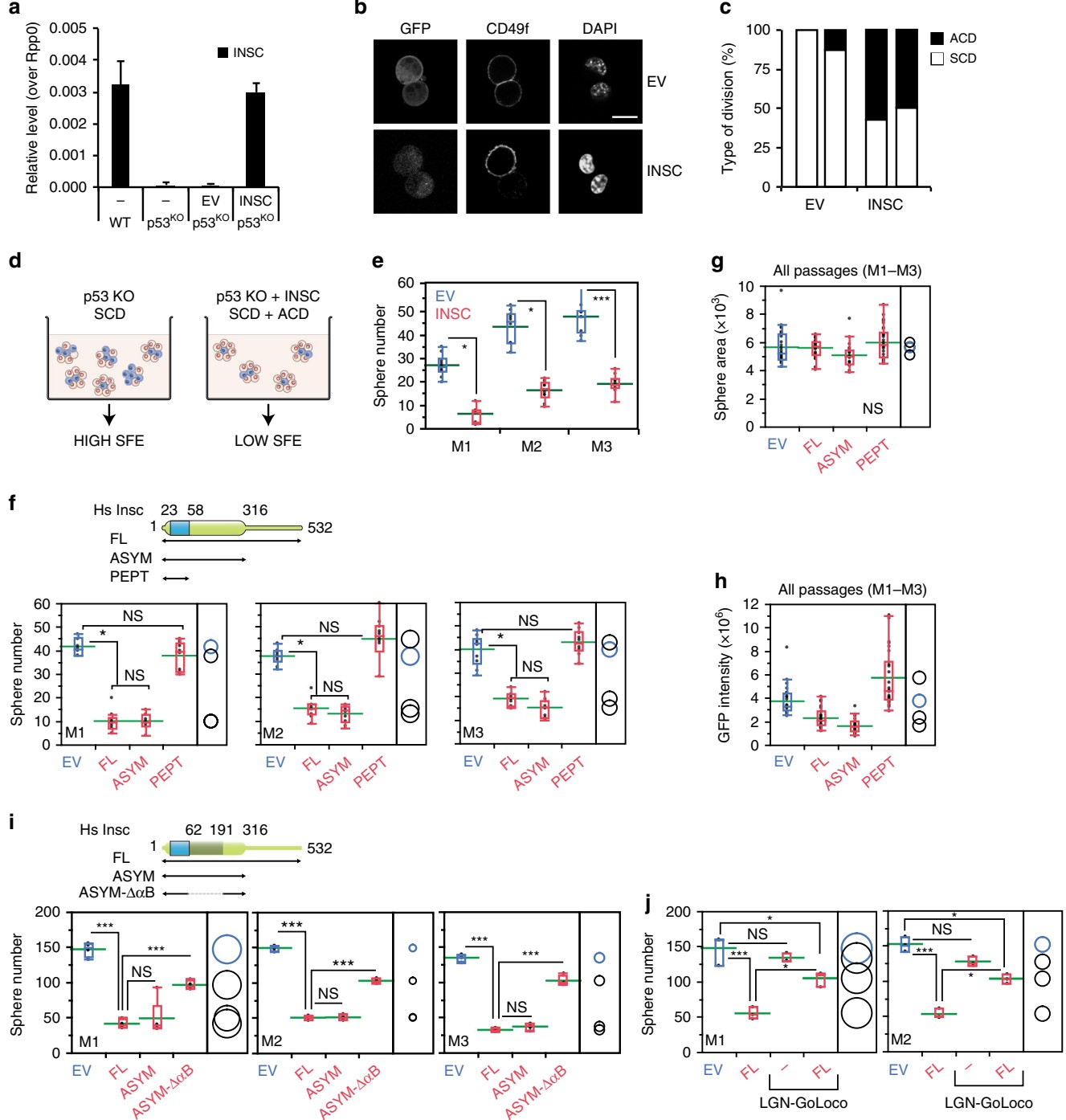

of fly and human assemblies could be ascribed to the longer human Insc full-length construct (about 58 kDa in size) as compared to the *Drosophila* Insc[ASYM] (about 33 kDa). To further corroborate the idea that human LGN and Insc form hetero-tetramers, we co-transfected HEK293T cells with two differentially tagged versions of LGN[TPR] (Flag-LGN[TPR] and GFP-LGN[TPR]) together with Insc[ASYM] or a variant of Insc[ASYM] lacking the region spanning residues 62–191 that corresponds to the four-helix bundle (hereon referred to as Insc[ASYM]-ΔαB). To improve detection in immunoblot, in this experiment we used GFP-tagged Insc[ASYM]-ΔαB. Flag-tagged human LGN[TPR] co-immunoprecipitates GFP- LGN[TPR] in the presence of Insc[ASYM], whereas in the presence of Insc[ASYM]-ΔαB it does not (Fig. 3f). Consistently, the LGN[TPR]:Insc[ASYM] complex elutes earlier from a SEC column than the LGN[TPR]:Insc[ASYM]-ΔαB complex (Supplementary Fig. 3g), confirming that Insc[ASYM]-ΔαB assemble in a lower stoichiometry complex than wild-type Insc. We conclude that (1) fly and human Insc[ASYM] are likely to share the same helical topology; (2) human Insc tetramerizes with LGN[TPR]; and (3) removal of the α-bundle results in the destablilization of the tetrameric assembly with a human Insc[ASYM] mutant associating with LGN[TPR] in a 1:1 stoichiometry.

**Organizational principles of the Par3:Insc:LGN:Gαi complexes.** Establishing apico-basal polarity is a prerequisite for asymmetric cell divisions. An evolutionary conserved function of Par3 has been described in marking the apical site of polarized mitoses, and conferring fate specification upon inheritance. In most instances Insc and LGN are found polarized with Par3[16,26,27]. In embryonic epidermis endogenous Insc and Par3 co-immunoprecipitate[3]. Similarly, human Insc and Par3 co-transfected in HEK293T cells co-immunoprecipitate via a direct interaction between the C terminus of Insc and the PDZ domains of Par3[23] (Supplementary Fig. 5a–c). Interestingly the whole PDZ domain of Par3 is required for binding to Insc (Supplementary Fig. 5d).

The identification of distinct regions of Insc binding to Par3 and LGN prompted us to ask whether Insc is the direct physical link between LGN and the polarity complex. Immunoprecipitation of LGN from mitotic lysates co-expressing Par3 in the presence or in the absence of Insc revealed that Insc is required for the association of LGN with Par3 (Fig. 4a). Consistently, the recombinant Par3[PDZ123] domain binds to the Insc:LGN:Gαi assembly purified from insect cells with a stoichiometry reflecting the amount of Insc, and elutes for a SEC column right after the 670 kDa marker, in agreement with the theoretical molecular weight of an oligomer containing two copies of Insc and LGN

(Fig. 4b). Collectively these findings demonstrate that the apical complex Par3:Insc:LGN:Gαi is held together by direct interactions centered on Insc:LGN tetramers, and connected to the plasma by Par3 and Gαi (Fig. 4c).

**Relevance of the Insc:LGN topology for breast stem cell ACD.** To assess the functional relevance of the Insc:LGN tetramers, we investigated the in vitro self-renewal of murine mammary stem cells (MaSCs) ectopically expressing Insc fragments designed on the basis of the structural analyses. Although cultured mammary epithelial cells may not fully reflect the properties of mammary stem cells residing in the mammary glands[15], we reasoned that in vitro analyses of isolated murine MaSCs could be used to dissect molecular mechanisms of cell division decoupled from niche signals. More specifically, previous studies reported that the first division of MaSCs isolated from wild-type mice is most frequently asymmetric, whereas the division mode of MaSCs derived from p53-KO mice is for the large majority symmetrical, eventually leading to the expansion of the SC compartment[28]. MaSCs from primary tissues of human and mouse origin can be grown in vitro in non-adherent conditions, giving rise to clonal spheroids called mammospheres[29]. In these assays, the number of mammospheres reflects the number of MaSCs originally present in the culture. MaSCs from the p53-KO mice expands aberrantly in culture as consequence of their symmetric mode of division, increasing the number of mammospheres observed over serial propagation[28,30]. Interestingly, qPCR analysis revealed that the Insc transcript is absent in p53-KO mammospheres (Fig. 5a), whereas Par3, LGN and NuMA are still transcribed (Supplementary Fig. 6a–c). Thus we reasoned that p53-KO MaSCs would be the ideal experimental setting to test whether Insc promotes asymmetry by reverting the symmetric division mode induced by p53 loss. We then infected p53-KO mammospheres with a lentiviral vector restoring the amount of full-length human Insc transcript at levels comparable to the endogenous one in wild-type MaSCs (Fig. 5a), and analyzed the division mode of MaSCs. Immunofluorescence analysis of doublets with the basal cell surface marker CD49f (ITGA6) showed that Insc-expressing MaSCs partition CD49f unequally (Fig. 5b, c), consistent with the role of Insc as determinant of ACD documented in other vertebrate stem cell systems[6,16,17]. Most importantly, in cultured MaSCs CD49f and Numb co-partition asymmetrically (Supplementary Fig. 6d), further corroborating the notion that in this system, ectopic expression of Insc promotes asymmetry. The increased proportion of ACDs versus SCDs observed in Insc-expressing p53-KO MaSCs predicts a reduction in their self-renewal capacity (Fig. 5d). To test this hypothesis, we set out to perform a serial propagation experiment using mammospheres

**Fig. 5** Insc[ASYM] induces ACDs in p53-KO mammary stem cells. **a** Insc transcript levels evaluated by RT−qPCR on mammospheres from wild-type FVB mice, or p53-KO mice infected with empty lentivirus (EV) or human full-length Insc (INSC) normalized to the Rpp0 control. Bars represent mean ± s.d. of three replicates. Statistics for 1 out of 2 independent experiments. **b**, **c** Images and quantification of the division mode of mammary epithelial cells from p53-KO mice evaluated by pair-cell assay. Cell infected with empty lentivirus (EV) or virus expressing full-length Insc (INSC) were stained with CD49f and DAPI. Scale bar, 10 μm. Quantification from 2 independent experiments with n > 10 cells for all conditions in each experiment. p = 0.005 according to the paired Student's test. **d** Scheme of the SFE assay used to evaluate self-renewal of p53-KO MaSCs. Stem cells are in blue, progenitors in pink. **e** Number of mammospheres from p53-KO mice infected with an empty lentivirus (EV) or a virus expressing Insc (INSC) counted over three serial passages and shown by box-plots with mean of 10 wells count per each passage (green line). **f**–**h** SFE assays with p53-KO mammospheres infected with empty virus (EV) or viruses expressing Insc (FL), Insc[ASYM] (aa 1–317, ASYM), or Insc[PEPT] (aa 1–58, PEPT). For each passage, the number of spheres (**f**), the sphere area (**g**) and the GFP intensity (**h**) have been evaluated for 10 wells, and plotted as in **e** (each pair Student's t-test). Statistics for 1 out of 3 independent experiments. **i** SFE assays performed with p53-KO mammospheres infected with empty lentivirus (EV) or viruses expressing Insc (FL), Insc[ASYM] (ASYM), or Insc[ASYM]-ΔαB (i.e. Insc[ASYM] lacking aa 62–191). For each passage, the number of spheres has been counted for four wells, and plotted as above (each pair Student's t-test). Statistics for one out of two independent experiments. **j** SFE assays performed with p53-KO mammospheres infected with empty lentivirus (EV) or viruses expressing Insc (FL), human LGN-GoLoco (aa 358–677) or both. For each passage, the number of spheres has been evaluated in three wells, and plotted as above (each pair Student's t-test). In the figure, *P < 0.01, ***P < 0.001 by Wilcoxon's test

from p53-KO mice infected with full-length Insc. Indeed, the sphere-forming efficiency (SFE) of p53-KO mammospheres infected with Insc displayed a 70% decrease, indicating a reduced clonogenic ability of the Insc-expressing MaSCs upon three serial passages (Fig. 5e). Notably, Insc ectopic over-expression does not perturb LGN, NuMA and Par3 levels (Supplementary Fig. 6a–c), indicating that the Par3:Insc:LGN complex and the spindle orientation motors works physiologically. To dissect the molecular requirements for the reduced self-renewal induced by Insc, we repeated the mammosphere assays expressing human Insc$^{A-SYM}$, spanning residues 1–317, which tetramerizes but is unable to bind Par3, or Insc$^{PEPT}$, sufficient for binding to LGN but not for oligomerization. SFE analyses revealed that comparable expression of Insc$^{ASYM}$ decreases self-renewal of p53-KO MaSCs to the same extent of full-length Insc, whereas Insc$^{PEPT}$ does not (Fig. 5f–h and Supplementary 6e, f). Mammospheres grown from MaSCs infected with different Insc constructs displayed the same surface area and GFP-reporter intensity, which remained unchanged upon the three analyzed passages, thus excluding proliferative defects or toxicity induced by lentiviral transduction (Fig. 5g, h). Importantly, inability to revert SCDs of p53-KO MaSCs in SFE assays was also observed upon lentiviral expression of the Insc$^{ASYM}$-ΔaB mutant forming 1:1 complexes with LGN (Fig. 5i and Supplementary 6g), further corroborating the notion that binding of Insc to LGN is not sufficient to promote ACDs, and that the teram=ric arrangement is key to mediate asymmetry.

In most of ACDs characterized so far, Insc polarizes with LGN at the apical site by direct interaction with the membrane-associated proteins Gαi and Par3[16], according to the topology that we characterized in the previous paragraph. To understand the relevance of Gαi in Insc-mediated ACDs, we conducted a SFE assay co-infecting p53-KO MaSCs with full-length Insc and a dominant-negative variant of LGN coding only for the C-terminal GoLoco region, which retains binding to Gαi but not to Insc, this way titrating away Insc:LGN complexes from the plasma membrane. In the presence of the LGN-GoLoco, Insc fails to reduce the sphere number in SFE assays (Fig. 5l and Supplementary Fig. 6h, i), demonstrating that the Gαi tethering of tetramers to the plasma membrane is a prerequisite for ACDs. Collectively, these data showed that restoring Insc levels in p53-KO MaSCs reverts symmetric divisions towards asymmetric ones and restricts self-renewal, and that the Insc$^{ASYM}$-mediated tetramerization of Insc with Gαi-bound LGN is essential for these functions.

The finding that Insc engages LGN in a stable tetramer, and that this interaction is competitive with the binding of LGN to the Dynein-adaptor NuMA[10,11], poses the question of how the Insc-bound and NuMA-bound pools of LGN relate to one another during ACDs. To gain a first insight into the relative abundance of LGN and Insc, we analyzed their transcript levels in human mammary epithelial CD44$^{high}$/CD24$^{low}$ cells isolated from HMLE cell line (immortalized human mammary epithelial cells), which exhibit properties of MaSCs[31]. Human mammary CD44$^{high}$/CD24$^{low}$ cells express significantly higher levels of Insc compared to the CD44$^{low}$/CD24$^{high}$ non-stem counterpart (about 2.5-fold) and LGN expression was far more abundant than Insc (Supplementary Fig. 6l). Although we cannot exclude that Insc and LGN protein levels are post-transcriptionally regulated, these data seem to suggest that Insc acts as limiting factor in the Insc:LGN complex formation. Because Insc forms stable tetramers with LGN, we can assume that NuMA is unable to prevent the formation and the activity of Insc:LGN tetramers in spite of competing with Insc for LGN binding and being more abundant. Conversely, since Insc is much less expressed than LGN, Insc:LGN complexes do not alter the spindle orientation functions

exerted by LGN in complex with NuMA/Dynein, but rather play a specific role that is essential for asymmetric stem cell divisions.

## Discussion

The current view posits that LGN contributes to asymmetric fate of oriented stem cell divisions by recruiting microtubule motors at the apical site via direct interaction with NuMA. Although LGN is clearly important to coordinate cortical polarity with mitotic spindle orientation, in this paper we show that a pool of LGN is engaged in a stable oligomeric assembly with Insc, Par3, and Gαi, which cannot exchange with NuMA. We speculate that this complex underpins a novel and conserved function of cortical Par3:Insc:LGN:Gαi complexes in promoting asymmetry by stabilizing fate defining components at the apical site, which is partitioned unequally upon cytokinesis.

Insc was originally identified in fly neuroblasts as an adaptor binding to dLGN. Previous structural studies showed that the short Insc$^{PEPT}$ (residues 304–340) binds to the dLGN$^{TPR}$ domain with nanomolar affinity, leaving open the issue as to why only a larger Insc fragment (termed Insc$^{ASYM}$) was required to recapitulate its functions in Insc-deficient neuroblasts[21,32]. The crystallographic structure of the dLGN$^{TPR}$:Insc$^{ASYM}$ complex reveals that Insc$^{ASYM}$ induces a stable dimerization of the dLGN$^{TPR}$ domains, which topologically prevents the dissociation of the two molecules. Together, these findings imply that the asymmetric function of the dLGN:Insc assembly is embedded in its oligomeric form.

In flies, the Insc$^{ASYM}$ domain develops C-terminally of the Insc$^{PEPT}$ in two helical domains, which have been previously assigned to specific Insc functions in neuroblasts. In particular, the fragment 302–459 roughly matching the helical bundle was shown to target Insc at the apical membrane[21], possibly because it is sufficient to trigger tetramerization with dLGN, while the Armadillo region encompassing residues 460–552 / 578 is required to drive asymmetric distributions of Numb and Miranda[21,32], for molecular reasons that still remain unclear. Based on in vitro reconstitution experiments, we cannot exclude that the Armadillo repeats contribute to the larger interface accounting for the interaction of dLGN$^{TPR}$:Insc$^{ASYM}$ with Par3$^{PDZ}$ and hence with the polarity complex (see below).

An important finding of our structural studies is that the functional homology between vertebrate and invertebrate Insc is rooted in the same overall organization of the assemblies that they form with their cognate TPR-GoLoco proteins. We found that the region of human Insc spanning residues 1–316 is predicted to fold as a helical domain resembling the fly Insc$^{ASYM}$, and that it forms analogous tetramers with human LGN$^{TPR}$. The evidence that in fly neuroblasts mouse LGN can rescue spindle defects induced by dLGN depletion corroborates the notion that their TPR domains assemble with Insc similarly[33].

Reconstitution of the whole human Par3:Insc:LGN:Gαi complex from recombinant sources showed that Insc links physically LGN to Par3 by direct binding to its PDZ domains. As expected for canonical PDZ ligands, the conserved C-terminal tail of Insc is the prominent determinant for the binding of Insc to Par3$^{PDZ123}$.

What is the relationship between the Insc-bound pool of LGN and the spindle orientation? The topology of the stable Insc:LGN tetramer implies that the LGN molecules engaged with Insc cannot be sequentially transferred to NuMA to pull on astral microtubules, as we had previously proposed[10,34]. Nonetheless, we show that Insc promotes asymmetric divisions in cultured p53-KO MaSCs, consistent with previous reports in skin progenitors and radial glial cells[4,6,16,17]. We speculate that only a minor fraction of LGN molecules associates with Insc during ACDs, leaving the bulk of the LGN population available for the

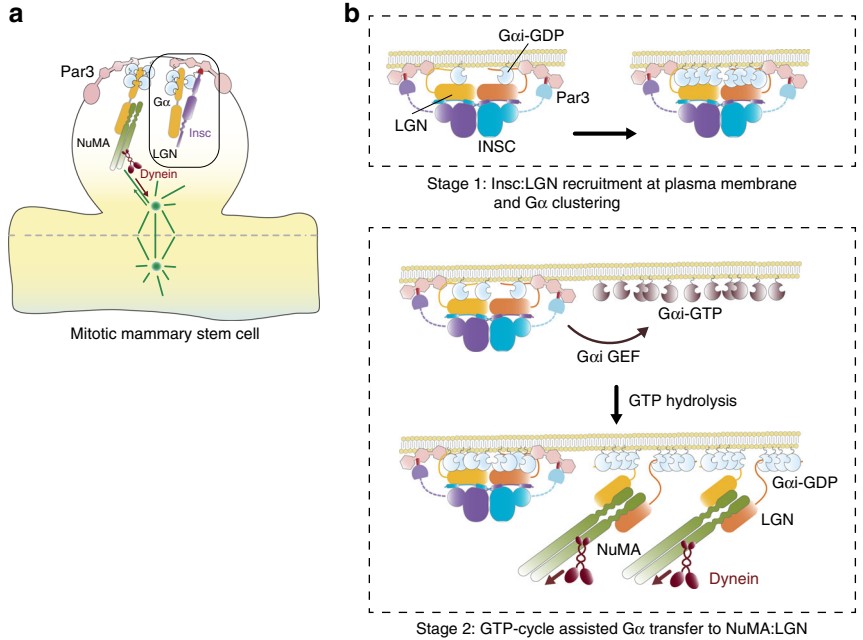

**Fig. 6** Model of the role of Insc:LGN complex in asymmetric divisions. **a** Schematic diagram depicting the organization of cortical domains and mitotic spindle motors in murine mitotic mammary stem cells. **b** Schematic representation of the asymmetric function of Insc:LGN tetramers supported by our study. Initally, Insc:LGN complexes are recruited at the apical membrane by direct binding of LGN to Gαi-GDP molecules, and by association of Insc with Par3. Cooperative binding to LGN-GoLoco motifs favors Gαi-GDP apical clustering. Later, a Gαi GTP-cycle, likely catalyzed by a GEF such as Ric-8A, dissociates Gαi from Insc:LGN, generating a localized pool of Gαi-GTP molecules, which upon GTP hydrolysis recruits LGN:NuMA:Dynein complexes at the apical site. According to this model, stable Insc:LGN assemblies couple asymmetric fate specification with apico-basal spindle orientation by Gαi-GDP transfer

assembly of NuMA/Dynein motor complexes. In this view, the two LGN-containing complexes would contribute to asymmetry in different ways: the Insc-bound pool by coordinating the distribution of fate determinants, including Par3, and the more abundant NuMA-bound pool by providing cortical information to orient the division plane (Fig. 6a). The evidence that fly Insc[ASYM] is involved in basal localization of Miranda and Pon/ Numb speaks in favor of this hypothesis.

Intriguingly, mammosphere assays revealed that the Par3-binding domain of Insc, residing outside the Insc[ASYM], is dispensible for ACDs, in agreement with previous findings in fly neuroblasts[21,32]. Conversely, binding of LGN to Gαi is essential to promote Insc-dependent MaSCs ACDs, suggesting that during ACDs Gαi likely acts as recruitment factor for Insc:LGN complexes at the apical membrane.

We speculate that the Gαi[GDP] moieties, possibly generated by specific GPCR activities[35], anchor Par3:Insc:LGN:Gαi complexes at a localized cortical site by direct binding to the LGN-GoLoco motifs. Because also NuMA:LGN assemblies contain Gαi[GDP], we think that GDP-loaded Gαi might be the common positional cue coupling asymmetric and orientation activities of LGN during ACDs, including the ones of cultured MaSCs, keratynocites and fly neuroblasts. In spite of competing with NuMA for LGN binding, during ACDs Insc is required for correct spindle orientation[6,9,16,18,26,36]. To explain this phenotype, we propose that stable Insc:LGN complexes are targeted at a restricted cortical site by direct interaction with Gαi[GDP], and favor clustering of Gαi by cooperative binding of each LGN-Goloco domain to four Gαi molecules. LGN associates only with the GDP-loaded form Gαi, thus it is plausible that specialized GEFs/GAPs catalyze a Gαi GTP-cycle that releases Gαi from Insc:LGN complexes and allow engagement of the same apically enriched Gαi molecules to LGN:NuMA, this way contributing to apico-basal spindle orientation (Fig. 6b). According to this model, aberrant boosting of

Insc levels in wild-type stem cells would eventually titrate away LGN from NuMA promoting misoriented symmetric divisions, as it was recently observed in MaSCs isolated from Insc knock-in mice[7]. The finding that ectopic over-expression of Insc in the developing skin results in a transient ACDs increase without morphogenetic long-term defects, indicate that feedback mechanisms exists to control the balance between ACDs and SCDs in tissues[17]. Further investigations will be needed to address the physiological relevance of our molecular findings for self-renewal of MaSCs residing in the mammary glands.

Finally, a very interesting observation stemming from our mechanistic studies is that Insc replenishment in p53-KO MaSCs reverts the symmetric division phenotype induced by p53 loss. Although the molecular basis underlying such remarkable effect are currently unclear, this finding indicates that Insc acts downstream of p53 in controlling MaSCs self-renewal. As the most aggressive breast tumors arise from deregulation of MaSCs homeostasis and self-renewal[37], and p53 is found mutated in nearly 40% of human breast cancers[28], we anticipate that experiments addressing the mechanisms underpinning the working principles of Insc in p53-KO MaSCs in vivo might be instrumental to the development of novel therapeutical strategies targeting breast cancer stem cells.

## Methods

**Protein preparation.** To generate a crystallization quality sample, the dLGN[25–444]: Insc[252–623] complex previously purified[10] was subjected to limited proteolysis, which resulted in two trimmed domains assigned by Mass Spectrometry to residues 25–406 of dLGN and residues 283–623 of *Drosophila* Insc. These two fragments were cloned into a dicistronic-modified version of the pGEX-6P1 vector (GE Healthcare), and co-expressed in BL21 Rosetta *E. coli* cells by induction with 0.5 mM IPTG overnight at 18 °C. Cells were lysed in 0.1 M Tris-HCl pH 8, 0.3 M NaCl, 10% glycerol, 0.5 mM EDTA, and 5 mM DTT, and cleared for 1 h at 100,000 *g*. Cleared lysates were affinity purified by incubation with Glutathione Sepharose-4 Fast-Flow beads (GE Healthcare). After washes, fusion proteins retained on beads were incubated with PreScission protease (GE Healthcare) overnight at 4 °C to

remove the GST-tag. The cleaved material was eluted from the beads in a desalting buffer consisting of 20 mM Tris-HCl pH 8, 40 mM NaCl, 5% glycerol, 5 mM DTT, and loaded on Resource-Q ion-exchange columns. Peak fractions were pooled, and loaded on a Superdex-200 column equilibrated in 10 mM HEPES pH 7.5, 0.6 M NaCl, 5 mM DTT. The collected fractions were quantified by absorbance reading at 280 nm, and concentrated up to 20 mg/ml. Human Insc:LGN:Gαi complexes were generated by co-infection of High5 insect cells with two baculoviruses and purified by affinity and anion exchange in the presence of Gαi obtained from bacterial sources[10]. The human Insc:LGN$^{TPR}$ complex was expressed in High5 insect cells infected with a recombinant dicistronic baculovirus vector coding for a 6His-tagged full-length Insc (residues 1–532, Uniprot Q1MX18-2) and the fragment 15–350 of LGN (Uniprot AAB40385.1). The complex was purified on NTA beads, followed by anion exchange chromatography. For pull-down and SEC experiments, constructs encompassing the three PDZ domains of human Par3 (residues 269–685, Uniprot Q8TEW0) was cloned in pGEX-6PI, expressed in BL21 Rosetta E.Coli, and purified by affinity followed by gel filtration chromatography after GST-tag removal.

**Crystallization and structure determination**. Initial crystals of the dLGN$^{TPR}$: Insc$^{ASYM}$ complex were obtained by sitting-drop vapor diffusion at 20 °C mixing 1 μl of the protein sample at 20 mg/ml with equal volume of a reservoir containing 20% PEG3350 and 0.2 M K-thyocianate. Optimization of the crystallization conditions resulted in plate-shaped crystals about 200 μm in size, using 1:1.5 volume ratio of protein to reservoir (13% PEG3350, 0.1 M Tris-HCl pH 7.8, 0.2 M K-thyocianate, 2.5% 1-Propanol, 2.5% Tacsimate pH 7.5) at 4 °C. For data collection, crystals were transferred to a cryo buffer (reservoir buffer supplemented with 25% glycerol), and flash-frozen in liquid nitrogen. X-ray diffraction datasets were collected to 3.4 Å and 4.0 Å resolution at I04 and I04-1 beamlines at Diamond Light Source. All data were processed with xia2[40]. Crystals belong to the space group P2₁2₁2₁. Initial phases were obtained by molecular replacement using the coordinate of dLGN$^{TPR}$:Insc$^{PEPT}$ (PDB ID 4A1S) as search model in Phaser[41]. Automatic model building performed on the different datasets using Buccaneer[42] yielded initial models, which were employed to perform multi-crystal average density modification in Phenix[43]. The quality of the resulting electron density maps allowed building of the complete model through iterative cycles of automatic and manual model building in Phenix and Coot[44] respectively. The final model was restrained refined to R$_{work}$ of 20.9% and R$_{free}$ of 24.9%, and good stereochemistry (Table 1). The final model contains eight copies of dLGN$^{TPR}$ (residues 39–392) and Insc$^{ASYM}$ (residues 306 to 595) assembled in four tetramers. The structure was visualized by Pymol (www.schrodinger.com/pymol) and Chimera[45].

**SAXS experiments**. SAXS data were collected using the SEC-SAXS setup at the ESRF BioSAXS beamline BM29 (Grenoble, France). 100 μl of the dLGN$^{TPR}$: Insc$^{ASYM}$ complex at a concentration of 20 mg/ml were injected on a Superdex-200 10/30 Increase column (GE Healthcare), and 1000 individual frames of 2 s exposure were collected on the eluate using the Pilatus 1 M detector (Dectris). Individual frames were processed automatically using an EDNA based processing pipeline. Frames for further processing were chosen by selecting frames according to the match with matching radius of gyration and high similarity using COR-MAP[38]. Guinier-analysis and calculation of the pair distribution function were done in Primus from the ATSAS package[46]. 20 ab initio models for both P1 and P2 symmetry were calculated using DAMMIF[47], and then averaged, aligned and compared using DAMAVER[48]. A representative bead model was calculated using DAMMIN[49], and aligned to the crystallographic structures using SUPCOMB and PyMol. Theoretical scattering curves of the known dLGN$^{TPR}$:Insc$^{ASYM}$ structures were calculated and fitted to the experimental curve using WAXSiS[50]. Details of SAXS data acquisition and processing are summarized in Supplementary Table 1.

**Immunoprecipitation experiments**. Human Insc (residues 1–532) and InscΔC (residues 1–516) were cloned into a modified version of pCDNA5 with a N-terminal HA-tag, human LGN (residues 1–677) was cloned into pCDNA5 with a C-terminal FLAG-tag, and Par3 was cloned into pEGFP_N1 (Clontech). HEK293T cells were cultured in Dulbecco's Modified Eagle's Medium (DMEM) supplemented with 10% of fetal bovine serum and 1% L-glutamine. Cells were transfected with the constructs indicated in the experiments, and after 48 h from transfection cells were treated with 3.3 μM of nocodazole (Sigma) for 16 h. Cell lysates prepared in JS buffer (50 mM HEPES pH 7.4, 0.15 M NaCl, 10% glycerol, 1% Triton, 1.5 mM MgCl₂, 5 mM EGTA and protease inhibitor cocktail) were incubated with 2 mg/ml of α-FLAG M2 antibody (Sigma) or α-GFP antibody (MBL) conjugated to sepharose beads for 2 h at 4 °C. Immunoprecipitated proteins were then washed four times in JS buffer, and analyzed by SDS–PAGE and immunoblotting.

Human Insc$^{ASYM}$ (residues 1–317), GFP-Insc$^{ASYM}$-ΔαB (corresponding to Insc$^{ASYM}$ lacking residues 62–191), GFP-LGN$^{TPR}$ and FLAG-LGN$^{TPR}$ (residues 1–350) were cloned into a pCDH vector with hygromycin resistance under a CMV promoter (CD515B System Biosciences). HEK293T cells cultured as described above were transfected with pCDH vectors expressing the different constructs as indicated in the experiments of Fig. 3f. After 48 h from transfection, cell lysates prepared in modified JS lysis buffer (50 mM HEPES pH 7.4, 0.15 M NaCl, 10%

glycerol, 0,5% NP40, 1.5 mM MgCl₂, 5 mM EGTA, 2 mM DTT and protease inhibitor cocktail) were incubated with 2 mg/ml of α-FLAG M2 antibody (Sigma) conjugated to sepharose beads for 1 h at 4 °C. Immunoprecipitated proteins were then washed three times in lysis buffer, separated by SDS–PAGE, and analyzed by immunoblotting.

**SEC of cell lysates**. For experiments of Supplementary Fig. 3g, HEK293T cells cultured as described above were transfected with pCDH vectors expressing GFP-Insc$^{ASYM}$ and FLAG-LGN$^{TPR}$ constructs (generated as described in the previous paragraph) as indicated in the experiments. After 48 h from transfection, cell lysates prepared in lysis buffer (50 mM HEPES pH 7.4, 0.15 M NaCl, 10% glycerol, 0,5% NP40, 5 mM EDTA, 5 mM DTT and protease inhibitor cocktail) were filtered (0.22 μm filter) and quantified. 0.5 ml of cell lysates at a concentration of 6 μg/μl were loaded on a Superdex-200 10/30 equilibrated in 50 mM HEPES pH 7.4, 0.15 M NaCl, 10% glycerol, and 5 mM DTT. Eluted fractions were collected, separated by SDS–PAGE, and analyzed by immunoblotting.

**Pull-down assays**. Bacterially expressed GST-tagged Par3$^{PDZ}$ (2 μM) was incubated with HEK293T cell lysates expressing Insc full-length or InscΔC for 2 h at 4 ° C in JS buffer. After four washes in JS buffer, proteins retained on beads were resolved by SDS–PAGE, and detected by immunoblotting. Membranes were then stained with Coomassie or Ponceau to visualize the amounts of GST-fusion proteins.

**Mammosphere cultures**. Mammary tissues from p53-KO mice were processed, and primary mammospheres obtained as described[28]. Briefly, mammary tissues were mechanically dissociated and placed in a digestion medium (DMEM supplemented with 200 U/ml collagenase (Sigma) and 100 U/ml hyaluronidase (Sigma)) for 3 h at 37 °C. Cells were plated onto ultra-low attachment plates (Falcon) at a density of 50,000 viable cell/ml in a serum-free mammary epithelial basal medium (MEBM, BioWhittaker), supplemented with 5 μg/ml insulin, 0.5 μg/ml hydrocortisone, 2% B27 (Invitrogen), 20 ng/ml EGF and bFGF (BD Biosciences), and 4 μg/ml heparin (Sigma). Mammospheres were collected after 7 days and mechanically dissociated. Cells were cultured in MEBM, 0.5 mg/ml hydrocortisone, 5 μg/ml insulin, 4 μg/ml heparin (Sigma), 20 ng/ml epidermal growth factor (EGF), 20 ng/ml fibroblast growth factor (βFGF), 1% Pen/Strep, 2 mM glutamine. Mammary cells were infected with pCDH-CMV-EF1-GFP lentiviral vectors coding for Insc constructs (Insc full-length, FL; the asymmetric domain spanning residues 1–317, ASYM; and the LGN-binding peptide encompassing residues 1–58, PEPT), and selected with 2 μg/ml of puromycin in solution. To analyze MaSCs self-renewal, cells from disaggregated mammospheres grown in liquid culture were either diluted at 2000 cells/ml for serial propagation, or plated in 96 multi-well plates (1000 cells per well) in a semisolid stem cell medium supplemented with methylcellulose. Disaggregation and re-plating at the same density were repeated for three passages, at a 7-day distance. For Fig. 5e–h, cell plating in methylcellulose was performed in 10 technical replicates, which were analyzed automatically using the JMP software (SAS) to quantify mammospheres numbers, GFP intensity and diameter. For these measurements, only spheres above a 100 μm diameter size threshold were considered. To monitor Insc protein levels, cell lysates from mammosphere cultures were prepared in JS and analyzed by Western blotting using anti-Inscuteable (rabbit polyclonal, raised against fragment 1–302 of human Insc, produced in house), and anti-vinculin antibody as normalizing control (mouse monoclonal, clone hVIN-1, Sigma). Protein bands were visualized using the SuperSignal West Pico Substrate (Pierce, Rockford, IL) after incubation with an HRP-conjugated anti-mouse secondary antibody (Sigma, St. Louis, MO). For Fig. 5i, mammary cells were infected with pCDH-CMV-EF1-GFP lentiviral vectors coding for Insc constructs (Insc full-length, FL; Insc$^{ASYM}$ spanning residues 1–317, ASYM; and Insc$^{ASYM}$-ΔαB corresponding to Insc$^{ASYM}$ lacking residues 62–191, ASYM-ΔαB). For experiment of Fig. 5l, mammary cells were infected with pCDH-CMV-hygro lentiviral vectors coding for mCherry or mCherry-LGN-GoLoco (residues 358–677). Cells were selected with 2 μg/mL of puromycin and 1 μg/ml hygromycin in solution. To analyze MaSCs self-renewal, cells from disaggregated mammospheres grown in liquid culture were either diluted at 2000 cell/ml for serial propagation, or plated in 24 multi-well plates (1000 cells per well) in a semisolid stem cell medium supplemented with methylcellulose. Disaggregation and re-plating at the same density were repeated for three or two passages, at a 7-day distance. Cell plating in methylcellulose was performed in four (Fig. 5i) or three (Fig. 5l) technical replicates. To monitor Insc protein levels, cell lysates from mammosphere cultures were prepared in JS buffer, and analyzed by western blotting using anti-Inscuteable antibody (rabbit polyclonal, raised against fragment 1–302 of human Insc, produced in house, working dilution 1:1,000), anti-mCherry (rat monoclonal, Thermo Fisher, M11217, working dilution 1:10,000)), and anti-vinculin antibody as normalizing control (mouse monoclonal, clone hVIN-1, Sigma, working dilution 1:10,000).

**Real-time PCR analysis**. Total RNA was extracted from mammary epithelial cells using Trizol (Invitrogen) and miRNeasy micro kit (QIAGEN), according to manufacturer's instructions. Reverse-transcription was performed using the Im-Prom II Reverse Transcriptase kit for cDNA synthesis (Promega). Real-time

quantitative PCR (RT–qPCR) was performed with the LightCycler480 System (Roche) with the primers listed in Supplementary Table 2. PCR efficiency of primer pairs were calculated using a standard curve to allow efficient quantification of expression and comparison of levels between different genes (Supplementary Fig. 6i). Notably, mmInsc primers recognizes both mouse and human Insc, and were used for RT–qPCR of Fig. 5a, while hsInsc primers recognizes only human Insc and were used to assess the expression levels of Insc mutants in Supplementary Fig. 6.

**Immunofluorescence**. For MaSCs doublet analyses, mammospheres were mechanically dissociated and single cells were allowed to divide once for 30 h in stem cell medium additioned with 20% methylcellulose in order to avoid aggregation. Cells were then transferred to a poly-lysine coated glass slide (Corning) and fixed with 4% paraformaldehyde (PFA) for 20 minutes at room temperature. Then, cells were blocked in donkey serum diluted 1:5 (20%) for 15 min, and incubated with anti-CD49f antibody (working dilution 1:250; clone GoH3, cat. no 555734, BD Pharmigen) for 1 h at room temperature. For Supplementary Fig. 6d, cells were further stained with a monoclonal Numb antibody as described in Colaluca et al.[51]. Cells were washed with PBS, and later incubated with anti-rat cy3 antibody (working dilution 1:400; Jackson ImmunoResearch) for 30 min at room temperature. Following that, cells were washed again with PBS, and fixed with 4% PFA for 10 min at room temperature. After permeabilization with 0,1% Triton-X, cells were counterstained with DAPI (Sigma). Confocal microscopy was performed on a Leica TCS SP2 confocal microscope. A 63 × oil-immersion objective lens (HCX Plan-Apochromat 63× NA 1.4 Lbd Bl; Leica) was used for analysis.

**Data availability**. Data are available on request from the authors. The coordinates and structure factors of the structure are available from the Protein Data Bank under the accession number 5A7D.

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

## Acknowledgements

We are grateful to Sebastiano Pasqualato, Valentina Cecatiello and Nicoletta Caridi of the IEO Crystallography Unit for technical support. We thank scientists at the I04 and I04-1 beamlines at Diamond, the X06DA beamline at the SLS, and the ID23-2 beamline at ESRF for precious help with data collection. We thank Fabiana Panebianco, Fernanda Ricci and Mark Wade for the help with mammosphere assays. We are grateful to all members of our laboratory for scientific discussions. This work is supported by the Italian Foundation for Cancer Research (AIRC) to M.M. (IG18692) and F.N. (IG14085), the Italian Ministry of Health to M.M. (RF-2013-02357254), and BioStruct-X (FP7/2007-2013, grant agreement N°283570) to M.M.

## Author contributions

S.M. conducted biochemical reconstitutions in vitro and experiments with MaSCs. S.C. determined and analyzed the structure. S.G. helped with IP experiments, G.B. helped with protein purification. P.B. and F.N. provided expertize and support for studies in MaSCs. M.B. and A.R. performed SAXS experiments. M.M. designed the study and wrote the manuscript.

## Additional information

**Competing interests:** The authors declare no competing interests.

