## [Peer Review File · Nature Communications]

Reviewer #3 (Remarks to the Author):

The authors nicely addressed to the majority of issues that I have raised in the previous review. Remaining concerns are as follows,

- 1, The Methods section seems to be missing in the revised manuscript.
- 2, The results of the statistical analysis of Figure 5c and Supple Fig. 6i should be described in Figure legends. Although the authors say that they have updated Figure 5c in the cover letter, the revised Figure 5c is the same as the previous version.

Reviewer #4 (Reviewer 1 replacement, Remarks to the Author):

This manuscript is concerned with the mechanism of mitotic spindle orientation, which is important in asymmetric stem cell divisions and epithelial morphogenesis. It provides a structure of the *Drosophila* Partner of Inscuteable (Pins) protein in a tetrameric complex with a domain of its partner, Inscuteable. This complex can interact with Par3 and the G- α -I subunit to form a stable unit that cannot be dissociated by NuMA. The authors propose that the interaction of this fragment of Inscuteable is sufficient to drive asymmetric cell division independently of Par3. The revisions that have been introduced in response to the comments of the initial reviews are for the most part appropriate and improve the manuscript. Overall, the structural section of the study seems sound, and describes a novel type of complex between Insc and Pins, which will be of interest to the field. However, I have some concerns about the biological analyses in the latter part of the manuscript, as described below:

1. The authors use RT-PCR to compare expression levels of Pins, Insc, NuMA and Par3, and claim that Insc is expressed at a lower level than Pins. However, one cannot extrapolate from mRNA levels to protein levels. There are, of course, multiple post-translational mechanisms that regulate protein levels, so as expected the correlation between mRNA and protein is very poor (see for example Gygi et al MCB 1999; Zhang et al, Nature 2014: Proteomic characterization of human colon and rectal cancer). For mRNAs of the same value protein levels can vary more than 20-fold, even in budding yeast. Therefore, these data do not provide any useful information on protein abundances and need to be removed.

2. In their new model (Figure 6), the authors suggest that rather than Par3 being involved in Insc/Pins clustering at the apical domain, G- α -i performs this function. However, where it has been studied, G- α -i does not show a polarized distribution at the plasma membrane (e.g., Du and Macara, Cell 2004), so the proposed mechanism does not seem plausible. Moreover, there is no evidence that the mammary "stem" cells are polarized in an apical/basal manner, and in fact this seems highly unlikely to be the case. In general, the authors seem to frequently confuse the *Drosophila* neuroblast model (in which the stem cell derives by delamination from the polarized neuroectoderm) with the much less-well characterized mammary cells they are using.

3. It is perhaps unfortunate that the authors chose cells from the mammary gland to test their model. Where careful, quantitative lineage-tracing techniques have been employed the weight of evidence is strongly against the existence of multipotent stem cells in the postnatal murine mammary gland (see, most recently, Wuidart et al Genes & Dev 2016). Despite the reference to a paper claiming that there are mammary stem cells in the terminal end buds that divide asymmetrically in an Insc-dependent manner (Ballard et al, Cell Rep 2015) this study did not employ lineage tracing and is based mostly on extrapolations from stained sections. A lineage tracing study using the SMA promoter (which is expressed in the end bud cap cells) found no evidence that these cells, in situ, function as bipotent stem cells, but are instead unipotent

progenitors of the basal lineage (Prater et al, Nature Cell Biol 2014). It is true that once these cells – or even differentiated basal cells - have been isolated from the mammary gland they gain stemness in vitro and exhibit multipotency. But CD49f, used here as a stem cell marker, is in fact a marker of the basal cell lineage. Whether these cells can really undergo asymmetric cell divisions after acquisition of stemness is still highly controversial.

4. Although loss of p53 has been reported to interfere with asymmetric cell divisions there are other reasons why loss of this checkpoint control protein could result in increased mammosphere growth – so it would be important to test whether Insc over-expression inhibits cell proliferation in general, which would give the same result as the one proposed in Figure 5. While massive over-expression of Insc might induce asymmetric cell division in these cells, the biological significance remains uncertain. (It is unclear how much Insc is over-expressed in the experiments shown in Figure 5). Therefore, at a minimum, I feel that the authors need to acknowledge the highly artificial nature of their cell-based experiments, and that the physiological relevance – if any - needs further study.

Reviewer #5 (Reviewer 2 replacement, Remarks to the Author):

The concerns of Review 2 have been partially addressed, and the authors still need to consolidate their manuscript before publication in Nature Communications.

1) The SEC elution peak of dLGNTPR/InscASYM complex in Fig. 3e is highly asymmetric, which means that the apparent peak could be an overlap of a tetramer peak (major portion) and a dimer peak (minor portion). Thus Review 2 proposes that a portion of the complex remains as a dimer. Also, the LGNTPR/mInsc complex seems to form larger oligomers in addition to tetramer (Fig. 3e). From the point-by-point response, the authors argue that "the higher molecular weight species represent a minor population of Insc:LGN complexes aggregating aspecifically with contaminants rather than genuine oligomers with stoichiometry higher than 2:2" (page 4, last sentence for responding point 1). However, no data was provided for their contaminant hypothesis. In addition, the elution profiles of Flag-LGNTPR cotransfected with GFP-/His-Insc WT or mutants also showed formation of larger oligomers (point-by-point response, page 8, panel b). It's really odd that the same "contaminants" could be observed several times in their experiments. I understand that the authors are trying to claim the importance of the tetramer formation, though their data is in contradiction to their statement that "the tetramer extremely stable" (page 6, line 129). I agree with Review 2 that the tetrameric state of the complex captured in the current study maybe one of many forms of the complex, though it is required for asymmetric cell division. I would suggest the authors to tune down their statement.

2) The model proposed by the authors that Par3 is not required for the apical enrichment of Insc-LGN, and Par3 acts downstream of Insc in ACD is not well supported by their own results. The authors need to focus their manuscript on the Insc/LGN tetramer, for which they have compelling evidence. The data and model related to the Par3-Insc interaction are limited, and not convincing. I suggest that the authors eliminate the Par3-Insc part, or it should be move to the supplementary section. Abstract, Results and Discussion need to be changed accordingly.

Reviewers' comments:

Reviewer #3 (Remarks to the Author):

The authors nicely addressed to the majority of issues that I have raised in the previous review. Remaining concerns are as follows,

- 1, The Methods section seems to be missing in the revised manuscript. We have provided the Methods in a separate file.

2, The results of the statistical analysis of Figure 5c and Supple Fig. 6i should be described in Figure legends. Although the authors say that they have updated Figure 5c in the cover letter, the revised Figure 5c is the same as the previous version.

We have added the description of the statistical analysis of Fig. 5C, which were previously mentioned only in the Rebuttal to Reviewer 1, to the corresponding legend. Suppl. Fig. 6i has been removed according to Reviewer 4's suggestions.

Reviewer #4 (Reviewer 1 replacement, Remarks to the Author):

This manuscript is concerned with the mechanism of mitotic spindle orientation, which is important in asymmetric stem cell divisions and epithelial morphogenesis. It provides a structure of the *Drosophila* Partner of Inscuteable (Pins) protein in a tetrameric complex with a domain of its partner, Inscuteable. This complex can interact with Par3 and the G-alpha-I subunit to form a stable unit that cannot be dissociated by NuMA. The authors propose that the interaction of this fragment of Inscuteable is sufficient to drive asymmetric cell division independently of Par3. The revisions that have been introduced in response to the comments of the initial reviews are for the most part appropriate and improve the manuscript. Overall, the structural section of the study seems sound, and describes a novel type of complex between Insc and Pins, which will be of interest to the field. However, I have some concerns about the biological analyses in the latter part of the manuscript, as described below:

1. The authors use RT-PCR to compare expression levels of Pins, Insc, NuMA and Par3, and claim that Insc is expressed at a lower level than Pins. However, one cannot extrapolate from mRNA levels to protein levels. There are, of course, multiple post-translational mechanisms that regulate protein levels, so as expected the correlation between mRNA and protein is very poor (see for example Gygi et al MCB 1999; Zhang et al, Nature 2014: Proteomic characterization of human colon and rectal cancer). For mRNAs of the same value protein levels can vary more than 20-fold, even in budding yeast. Therefore, these data do not provide any useful information on protein abundances and need to be removed.

We agree with the Reviewer that extrapolating protein levels from mRNA levels might not be necessarily correct. Unfortunately measuring relative protein levels in wild-type MaSCs was out of reach for us. Therefore, to have a first insight into the cellular amounts of LGN, Insc, NuMA and Par3, we decided to evaluate the relative transcript levels, which seemed to support the idea that Insc is less expressed than the other components of this protein network. However, given the limitation of this analysis, we have removed the data of Suppl. Fig. 6i and Suppl. Fig S7, as suggested by the Reviewer. In addition, to avoid any additional confusion, we have clearly stated in the manuscript where *transcript levels* (not protein expression levels) have been analysed at pg. 11, 13 and 19.

2. In their new model (Figure 6), the authors suggest that rather than Par3 being involved in Insc/Pins clustering at the apical domain, G- α -i performs this function. However, where it has been studied, G- α -i does not show a polarized distribution at the plasma membrane (e.g., Du and Macara, Cell 2004), so the proposed mechanism does not seem plausible. Moreover, there is no evidence that the mammary “stem” cells are polarized in an apical/basal manner, and in fact this seems highly unlikely to be the case. In general, the authors seem to frequently confuse the *Drosophila* neuroblast model (in which the stem cell derives by delamination from the polarized neuroectoderm) with the much less-well characterized mammary cells they are using.

We thank the Reviewer for the comments. It is true that we do not have experimental evidence to support the idea that the GDP-bound pool of G α i that in mitosis is engaged with LGN is clustered above the spindle poles in cultured MaSCs. However, in most of known systems, in metaphase endogenous G α i is found in crescents above both spindle poles for symmetric divisions such as the ones of HeLa cells (Kiyomitsu & Cheeseman, NCB, 2012 in Supplementary Fig. S1f and S2c), or above one of the spindle poles for symmetrically dividing cells such as keratinocytes during development (Williams et al, NCB 2015 in Fig. 5b and 7a). We suspect that the homogeneous cortical distribution observed for YFP-G α i in Du and Macara (Du and Macara, Cell, 2004) might be driven by high over-expression levels.

We totally agree with the Referee that it is not known whether MaSCs have apico-basal epithelial polarization. What we wanted to illustrate in the cartoon of Fig. 6a is how the proteins described in our studies most likely distributes in metaphase cells in culture. More specifically, the main message we wanted to convey with the panel is that the LGN-bound pool of G α i-GDP is found in two distinct complexes, one with Insc and the other with NuMA, that likely localize in a crescent at the cortex. Confocal images from Ballard and co-workers (Ballard et al, Cell Rep 2015, Fig. 6a) indicate that LGN and NuMA follow this distribution in mammary epithelial cells. To clarify that in Fig. 6a we want only to represent the localization of LGN:G α i-GDP complexes, we have removed from the cartoon all polarity proteins beside Par3, for which we provided evidence that binds to Insc, and deleted the indication of the basement membrane, which might erroneously hint at an apico-basal polarization of the mammary epithelial cell. We have also indicated that this configuration is what we think happens at metaphase.

3. It is perhaps unfortunate that the authors chose cells from the mammary gland to test their model. Where careful, quantitative lineage-tracing techniques have been employed the weight of evidence is strongly against the existence of multipotent stem cells in the postnatal murine mammary gland (see, most recently, Wuidart et al Genes & Dev 2016). Despite the reference to a paper claiming that there are mammary stem cells in the terminal end buds that divide asymmetrically in an Insc-dependent manner (Ballard et al, Cell Rep 2015) this study did not employ lineage tracing and is based mostly on extrapolations from stained sections. A lineage tracing study using the SMA promoter (which is expressed in the end bud cap cells) found no evidence that these cells, in situ, function as bipotent stem cells, but are instead unipotent progenitors of the basal lineage (Prater et al, Nature Cell Biol 2014). It is true that once these cells – or even differentiated basal cells – have been isolated from the mammary gland they gain stemness in vitro and exhibit multipotency. But CD49f, used here as a stem cell marker, is in fact

a marker of the basal cell lineage. Whether these cells can really undergo asymmetric cell divisions after acquisition of stemness is still highly controversial.

We thank the Reviewer for the comments on the discussion regarding the existence of multipotent mammary stem cells in the mammary glands of adult mice. We are aware that the issue is controversial, and did not want to address multipotency with our *in vitro* studies. As reported by the Reviewer, mammary epithelial cells isolated from mammary glands and cultured *in vitro* exhibit stem properties and multipotency, and this is the ground for our analyses, which aimed at dissecting the role of Insc and its mutants designed on the basis of the structural data in a stem-like system *in vitro*. Due to the intrinsic difficulties in synchronizing and staining cells in mitosis with Insc or Par3, we decided to perform IF on doublets after the first division using the basal marker CD49f, for which we had well-working protocols (we have corrected the phrasing used to indicate CD49f from ‘stem cell marker’ to ‘basal cell marker’ at pg. 11). We have evidence that CD49f asymmetric partitioning correlates with asymmetric partitioning of Numb, which often contributes to fate specification. We have added these data in a new supplementary figure S6d in order to support the idea that Insc over-expression in p53-KO MaSCs induce the asymmetric inheritance of cellular components specifying cell identity, which in some respect might be taken as an indication of the existence of asymmetric mitoses *in vitro*.

4. Although loss of p53 has been reported to interfere with asymmetric cell divisions there are other reasons why loss of this checkpoint control protein could result in increased mammosphere growth – so it would be important to test whether Insc over-expression inhibits cell proliferation in general, which would give the same result as the one proposed in Figure 5. While massive over-expression of Insc might induce asymmetric cell division in these cells, the biological significance remains uncertain. (It is unclear how much Insc is over-expressed in the experiments shown in Figure 5). Therefore, at a minimum, I feel that the authors need to acknowledge the highly artificial nature of their cell-based experiments, and that the physiological relevance – if any - needs further study.

We thank the Reviewer for the comment. We fully agree that the mammosphere efficiency might not only reflect the proportion of symmetric versus asymmetric divisions, but also be affected by proliferative defects. To rule out the possibility that Insc expression in p53-KO MaSCs could alter the proliferation rates, we measured the sphere size after each plating and showed that it does not change upon expression of Insc constructs, as shown in Fig. 5g. In addition, we did not observe any difference in the cell number when we dissociated the mammospheres at consecutive passages for re-plating (data not shown). Thus, we think it is unlikely that Insc affects MaSC proliferation in our experiments.

Of note, based on the Insc transcript analyses shown in Fig. 5a, the expression levels of Insc ectopically expressed in p53-KO MaSCs (where Insc is absent) are comparable with the expression levels of Insc in wild-type MaSCs. Although we know that mRNA quantifications only provide a rough indication of the protein amount, these data exclude a massive overexpression of the protein. As discussed in the previous point of the response to the Reviewer,

due to posttranslational modifications or processing, it is still possible that in the infected mammary cells, Insc protein levels differ from the endogenous ones.

For all the reasons discussed in the above paragraphs, we agree with the Reviewer that our studies elucidate the molecular mechanism whereby Insc works in conjunction with LGN in mammary stem cells in culture, this way providing molecular guidelines to dissect the behaviour of murine mammary stem cells *in vivo* in future experiments. To fully clarify this point, have better explained the limit of our *in vitro* analysis in the introduction, throughout the text and in an explanatory sentence at pg. 10.

Reviewer #5 (Reviewer 2 replacement, Remarks to the Author):

The concerns of Review 2 have been partially addressed, and the authors still need to consolidate their manuscript before publication in Nature Communications.

1) The SEC elution peak of dLGNTPR/InscASYM complex in Fig. 3e is highly asymmetric, which means that the apparent peak could be an overlap of a tetramer peak (major portion) and a dimer peak (minor portion). Thus Review 2 proposes that a portion of the complex remains as a dimer. Also, the LGNTPR/mInsc complex seems to form larger oligomers in addition to tetramer (Fig. 3e). From the point-by-point response, the authors argue that "the higher molecular weight species represent a minor population of Insc:LGN complexes aggregating aspecifically with contaminants rather than genuine oligomers with stoichiometry higher than 2:2" (page 4, last sentence for responding point 1). However, no data was provided for their contaminant hypothesis. In addition, the elution profiles of Flag-LGNTPR cotransfected with GFP-/His-Insc WT or mutants also showed formation of larger oligomers (point-by-point response, page 8, panel b). It's really odd that the same "contaminants" could be observed several times in their experiments. I understand that the authors are trying to claim the importance of the tetramer formation, though their data is in contradiction to their statement that "the tetramer extremely stable" (page 6, line 129). I agree with Review 2 that the tetrameric state of the complex captured in the current study maybe one of many forms of the complex, though it is required for asymmetric cell division. I would suggest the authors to tune down their statement.

We thank the Reviewer for discussing the issue of the LGN:Insc stoichiometry of the complexes reconstituted with fly and human proteins. We agree that the SEC elution profile of *Drosophila* Pins-TPR:Insc-ASYM presented in Fig. 3e shows a pronounced tail, which we reasoned might be due to a mild excess of Pins-TPR in the prep used for this experiment. In fact, the SLS analysis presented in Fig. 3d and the SEC-SAXS analysis presented in Fig. 2 are fully consistent with a *Drosophila* Pins-TPR:Insc-ASYM monodispersed 2:2 complex in solution. Most notably, the Radius-of-giration (Rg) of the SEC-SAXS curve of Suppl. Fig. S3f is stable through-out the whole peak, indicating that the majority of the sample has the same 2:2 stoichiometry.

Unfortunately we cannot purify to homogeneity from insect cells the human LGN-TPR:Insc complex used for the SEC experiment depicted in Fig. 3e. The high-molecular weight contaminants which we refer to in the previous rebuttal letter are shown in the figure enclosed here below, corresponding to the uncropped human LGN-TPR:Insc SDS-PAGE of Fig. 3e.

The *contaminants* in the immuno-blots of the SEC runs of Flag-LGN-TPR cotransfected with GFP-/His-Insc reported in the previous point-by-point response to Referees are different because these complexes are not purified to homogeneity, they contain additional GFP molecules that increase the molecular weight of the complexes, and might contain other interactors. Thus, they are not comparable with the profiles of Fig. 3e and present broader peaks, which do not necessarily imply the existence of high-order oligomers. That said, we agree with the Reviewer that our characterisation of the stoichiometry of the human LGN:Insc complex is less accurate than the one reconstituted with *Drosophila* proteins. For this reason, we have softened our statement, and removed the sentence "the tetramer extremely stable" (page 6, line 129), as suggested by the Reviewer.

Superdex-200 - Figure 3e

Uncropped Coomassie-stained SDS-PAGE corresponding to the SEC elution profile of the human LGN-TPR:Insc complex purified from insect cells and presented in Fig. 3e. High molecular-weights contaminants are visible in the first lanes.

2) The model proposed by the authors that Par3 is not required for the apical enrichment of Insc-LGN, and Par3 acts downstream of Insc in ACD is not well supported by their own results. The authors need to focus their manuscript on the Insc/LGN tetramer, for which they have compelling evidence. The data and model related to the Par3-Insc interaction are limited, and not convincing. I suggest that the authors eliminate the Par3-Insc part, or it should be move to the supplementary section. Abstract, Results and Discussion need to be changed accordingly.

We agree with the Reviewer that the focus of our studies is the role of the Insc:LGN tetramers in asymmetric divisions, and that we do not present functional data for the importance of the Insc:Par3 interaction in this process. However, we believe that the data currently presented in Fig. 4 describing the reconstitution of the Par3:Insc:LGN complex in lysates and from recombinant sources are robust and novel. Therefore, if possible, we would prefer to keep them in a main figure. As suggested by the Reviewer, we have shortened the description of the direct binding between the two proteins at page 10 of the main text. In addition, to account for the lack of functional information regarding the role of Par3:Insc in cells and the sequence of events contributing to the assembly of functional cortical Par3:Insc:LGN:Gai complexes, we have rephrased the abstract, the results and the discussion, as asked by the Reviewer and revised the model presented in Fig. 6b, which depicts Par3 and Insc bound to one another throughout mitosis.

Reviewer #4 (Remarks to the Author):

Overall, the authors have addressed the main issues raised in the previous review. There are only 2 minor issues:

1. The authors refer to a new Supplementary Figure 6 but no such figure was included in the PDF of the supplementary information
2. Although the authors refer to an asymmetric distribution of Galphai in dividing HeLa cells from Kiyomitsu and Cheeseman the asymmetry is very small and in fact, in the text, these authors state: "NuMA and Gai1 also exhibited symmetric cortical localization, although Gai1 showed more homogeneous localization". The distribution was not quantified. I could not locate the NCB paper by Williams (2015).

Reviewer #5 (Remarks to the Author):

The authors have addressed all my concerns.

Department of Experimental Oncology
European Institute of Oncology
Via Adamello 16, Milan
I-20139, Italy

Marina Mapelli, PhD
+39.02.94375018
marina.mapelli@ieo.it

Ana Mateus
Associate Editor, *Nature Communications*
The Macmillan Campus, 4 Crinan Street
London N1 9XW
United Kingdom

January 10th, 2018

Dear Dr. Mateus,

please find enclosed the response to the Reviewers' comments to the manuscript 'Insc:LGN tetramers promote asymmetric divisions of mammary stem cells' which was assessed by Nat. Comm. Referees (NCOMMS-17-17538-T).

I thank you for the editorial help, and I look forward to hearing from you.

With my kindest regards,
Marina Mapelli

REVIEWERS' COMMENTS:

Reviewer #4 (Remarks to the Author):

Overall, the authors have addressed the main issues raised in the previous review. There are only 2 minor issues:

1. The authors refer to a new Supplementary Figure 6 but no such figure was included in the PDF of the supplementary information.

We apologize for the inconvenient. We made sure of adding the Supplementary Figure 6 in the final submission, and for the sake of completeness we included the supplementary figure and legend also here below.

Figure S6 - Mapelli

Supplementary Figure 6. ACDs of p53-null murine MaSCs require tetrameric assemblies of *Insc*^{ASYM} with *Gai*-bound LGN, but not Par3. (a-b-c) NuMA, Par3 and LGN transcript levels evaluated by RT-qPCR on mammospheres isolated from wild-type FVB mice, or p53-KO mice, later infected with an empty lentivirus (EV, control) or with a human full-length *Insc* (INSC) expressing lentivirus. Transcript levels were normalized to the *Rpp0* internal control. Bars represent mean \pm s.d. of three replicates. (d) Representative immunofluorescence images of

cultured MaSCs after the first division showing the co-localization of the basal marker CD49f and the fate determinant Numb. The distribution of Numb and CD49f were used to monitor ACD and to discriminate MaSCs from progenitor daughter cells. DNA is visualized with DAPI. Scale bar, 10 μ m. **(e-f)** Transcripts and protein expression levels of Insc constructs (full-length, FL; asymmetric domain, ASYM; and LGN-binding peptide, PEPT) lentivirally transduced in mammospheres obtained from p53-KO mice evaluated by RT-qPCR and by immunoblot. **(g)** Transcript levels of Insc constructs (full-length, FL; asymmetric domain, ASYM; and ASYM- $\Delta\alpha$ B) lentivirally transduced in mammospheres obtained from p53-KO mice evaluated by RT-qPCR. **(h)** Protein expression levels of Insc full-length (FL) with mCherry and mCherry-LGN-GoLoco lentivirally transduced in mammospheres obtained from p53-KO mice evaluated by immunoblot with the indicated antibodies. SFE assays conducted with these lentivirally transduced mammospheres are presented in Fig. 5I. **(i)** Representative images of mammospheres lentivirally transduced with mCherry and pCDH vectors expressing Insc and a GFP reporter. In each panel, white bars correspond to 400 μ m. **(l)** Expression levels of LGN and Insc in HMLE human mammary epithelial cells separated in CD44^{high}/CD24^{low} cells and their non-stem counterpart CD44^{low}/CD24^{high} cells. CD44^{high}/CD24^{low} and CD44^{low}/CD24^{high} cells were isolated by FACS-sorting as previously described³³. Data shows average and s.e.m. of three independent biological replicates, and are reported as Fragments Per Kilobase of exon per million fragments Mapped (FPKM). The Insc transcript is present only in the stem-like CD44^{high}/CD24^{low} population at levels significantly lower than LGN.

2. Although the authors refer to an asymmetric distribution of Galphai in dividing HeLa cells from Kiyomitsu and Cheeseman the asymmetry is very small and in fact, in the text, these authors state: "NuMA and Gai1 also exhibited symmetric cortical localization, although Gai1 showed more homogeneous localization". The distribution was not quantified. I could not locate the NCB paper by Williams (2015).

The Referee is right, we mis-cited the article. The article we referred to in our previous response is Williams *et al.* "Par3-mInsc and Gai3 cooperate to promote oriented epidermal cell divisions through LGN" NCB, 2014. DOI:[10.1038/ncb3001](https://doi.org/10.1038/ncb3001)

Reviewer #5 (Remarks to the Author):

The authors have addressed all my concerns.